# Elevational Variation in and Environmental Determinants of Fungal Diversity in Forest Ecosystems of Korean Peninsula

**DOI:** 10.3390/jof10080556

**Published:** 2024-08-07

**Authors:** Lei Chen, Zhi Yu, Mengchen Zhao, Dorsaf Kerfahi, Nan Li, Lingling Shi, Xiwu Qi, Chang-Bae Lee, Ke Dong, Hae-In Lee, Sang-Seob Lee

**Affiliations:** 1Department of Integrative Biotechnology, Sungkyunkwan University, Suwon 16419, Republic of Korea; 2Department of Environmental Health Sciences, Graduate School of Public Health, Seoul National University, Seoul 08826, Republic of Korea; 3Department of Biological Sciences, School of Natural Sciences, Keimyung University, Daegu 42601, Republic of Korea; 4Key Laboratory of Climate, Resources and Environment in Continental Shelf Sea and Deep Sea of Department of Education of Guangdong Province, Department of Oceanography, Key Laboratory for Coastal Ocean Variation and Disaster Prediction, College of Ocean and Meteorology, Guangdong Ocean University, Zhanjiang 524091, China; 5Department of Geosciences, Geo-Biosphere Interactions, Faculty of Sciences, University of Tuebingen, 72074 Tuebingen, Germany; 6Jiangsu Key Laboratory for the Research and Utilization of Plant Resources, Institute of Botany, Jiangsu Province and Chinese Academy of Sciences, Nanjing 210014, China; 7Biodiversity and Ecosystem Functioning Major, Department of Climate Technology Convergence, Forest Carbon Graduate School, Kookmin University, Seoul 02707, Republic of Korea; 8Department of Forestry, Environment and Systems, College of Science and Technology, Kookmin University, Seoul 02707, Republic of Korea; 9Department of Biological Sciences, Kyonggi University, Suwon 16227, Republic of Korea

**Keywords:** fungi, diversity patterns, elevation, environmental determinants

## Abstract

Exploring species diversity along elevational gradients is important for understanding the underlying mechanisms. Our study focused on analyzing the species diversity of fungal communities and their subcommunities at different trophic and taxonomic levels across three high mountains of the Korean Peninsula, each situated in a different climatic zone. Using high-throughput sequencing, we aimed to assess fungal diversity patterns and investigate the primary environmental factors influencing fungal diversity. Our results indicate that soil fungal diversity exhibits different elevational distribution patterns on different mountains, highlighting the combined effects of climate, soil properties, and geographic topology. Notably, the total and available phosphorus contents in the soil emerged as key determinants in explaining the differences in diversity attributed to soil properties. Despite the varied responses of fungal diversity to elevational gradients among different trophic guilds and taxonomic levels, their primary environmental determinants remained remarkably consistent. In particular, total and available phosphorus contents showed significant correlations with the diversity of the majority of the trophic guilds and taxonomic levels. Our study reveals the absence of a uniform diversity pattern along elevational gradients, underscoring the general sensitivity of fungi to soil conditions. By enriching our understanding of fungal diversity dynamics, this research enhances our comprehension of the formation and maintenance of elevational fungal diversity and the response of microbial communities in mountain ecosystems to climate change. This study provides valuable insights for future ecological studies of similar biotic communities.

## 1. Introduction

Ecologists have long endeavored to decipher biodiversity patterns, with variations along elevational gradients being pivotal in shaping the discourse on biodiversity patterns. The study of elevational diversity gradients represents a compelling facet of ecological research that reflects the complex interplay between abiotic and biotic factors that influence species distribution and abundance [1,2]. Numerous studies have documented the influence of elevation on a diverse array of fauna and flora [3,4], and the biodiversity of these organisms typically adheres to one of three general elevational patterns: a hump-shaped pattern with peak diversity at mid-elevations [5,6], monotonically decreasing diversity with increasing elevation [7,8], or monotonically increasing diversity with increasing elevation [9,10,11]. These patterns have been extensively studied in macroorganisms, providing substantial insights into their biogeographical distribution. However, the patterns exhibited by microorganisms along elevation gradients are more challenging to observe and measure compared to larger organisms in terrestrial ecosystems [12,13,14]. Therefore, exploring the distribution of microorganisms is essential to fully understand the nuances of biodiversity patterns along elevational gradients.

Fungi are crucial components of soil microbial communities and drive numerous vital ecological processes, including litter decomposition, nutrient cycling, and the regulation of plant growth [15]. Their presence and activity are pivotal for maintaining the health and stability of ecosystems, thereby influencing both aboveground and belowground biodiversity [16,17,18]. Emerging research has revealed diverse fungal distribution patterns in terrestrial ecosystems [19,20,21,22]. Fungal communities are often classified into various taxonomic groups and functional guilds [23]. For example, mycorrhizal fungi enhance plant nutrient uptake, whereas saprotrophic fungi play a key role in organic matter decomposition [24,25]. The diversity of these fungal subcategories tends to change along elevational gradients. For example, studies have shown that the diversity of arbuscular mycorrhizal fungi is negatively correlated with elevation [26]. Nevertheless, when studying the entire fungal community, or its different taxonomic groups and functional guilds, these fungi consistently demonstrate rapid responses to environmental changes.

The diversity of soil fungal communities along elevational gradients is influenced by the complex interplay between various environmental factors. Many studies have emphasized the substantial impact of climatic conditions, such as mean annual temperature (MAT) and mean annual precipitation (MAP), on fungal diversity [21,27]. Soil properties, such as the soil pH and C:N ratio, are also recognized as factors that affect the diversity of fungal communities [28,29,30]. Additionally, geographical factors play a crucial role in shaping fungal diversity [31]. These findings illustrate the complex interactions among the environmental factors that regulate soil fungal diversity. However, despite extensive research on this topic, a lack of consensus remains regarding the response of fungal diversity to changes in environmental factors. By understanding how fungal community diversity responds to changes in climatic factors, soil properties, and geographical factors, we can predict how ecosystems will change under varying climatic conditions. This knowledge is essential for developing strategies to adapt to and mitigate the impacts of environmental change.

To gain a deeper understanding of the patterns of soil fungal diversity along elevational gradients, as well as the environmental factors influencing this diversity, we collected soil samples at six different elevations on three mountains in the Korean Peninsula. By sequencing the internal transcribed spacer (ITS) region, we investigated the fungal communities on these mountains with the objectives of clarifying (1) the patterns of fungal diversity along the elevational gradients on these three mountains, (2) the relationship between fungal diversity and environmental factors, and (3) the associations between elevational and environmental gradients and fungal diversity at different taxonomic or functional levels. This study provides a comprehensive examination of how fungal communities respond to elevational and environmental gradients, thereby contributing valuable insights into the biogeographic patterns of fungal diversity.

## 2. Materials and Methods

### 2.1. Sampling Site

Sampling was conducted in August and September of 2019 and 2020 and focused on three prominent mountains in the East Asian temperate zone: Mt. Halla, Mt. Jiri, and Mt. Seorak (Figure 1). These mountains were selected for their distinct climatic and ecological features that are representative of the region’s biodiversity.

Mt. Halla (33.3° N, 126.5° E), located on Jeju Island, is the highest shield volcano in South Korea, with an elevation of 1950 m. Jeju Island has a warm temperate climate with an MAT of 15.3 °C and MAP of 1424 mm. The MAT at the summit of Mt. Halla is considerably lower, reaching only 3.7 °C. The native vegetation on the lower slopes of Mt. Halla mainly consists of evergreen broadleaf forests dominated by communities of *Castanopsis cuspidata* var. *sieboldii*, *Quercus salicina*, and *Quercus glauca*. As the elevation increases, the predominant vegetation transitions from deciduous *Quercus serrata* forests (600–1400 m) with *Abies koreana* (1400 m) to shrub belts that include *Juniperus chinensis*, *Empetrum nigrum*, and *Ilex crenata* [12,32].

Mt. Jiri (35.3° N, 127.7° E), standing at 1915 m, is the second highest peak in South Korea. It has a MAT of 13 °C and MAP of approximately 1400 mm [33]. Low-elevation areas are dominated by *Quercus variabilis* and *Pinus densiflora*, while mid-elevation zones feature a mix of *Carpinus laxiflora*, *Quercus serrata*, *Acer mono*, *Larix leptolepis*, and *Pinus koraiensis* (Choi and An, 2013). Higher elevations are characterized by *Quercus mongolica*, *Agrostis clavata*, *Rhododendron mucronulatum*, and *Abies koreana* [34].

Mt. Seorak (38.08° N, 128.3° E), located in Eastern South Korea, has an elevation of 1708 m. It has the lowest MAT among the three mountains, averaging 3.05 °C along its elevation gradient, and a MAP of 1537 mm [35]. The diverse vegetation includes low-elevation species, such as *Pinus densiflora* and *Abies holophylla* in the lowlands and *Betula ermanii*, *Pinus koraiensis*, *Quercus mongolica*, and *Abies nephrolepis* in the highlands. The summit and highlands are home to dwarf species, such as *Pinus pumila*, *Taxus caespitosa*, and *Thuja koraiensis*, as well as arctic–alpine plants, such as *Arctous ruber*, *Crataegus komarovii*, and *Vaccinium uliginosum* [36].

The sampling methods were consistent across the three mountains. Soil samples were collected from six different elevations along an elevational transect, with five sampling points at each elevation band, and approximately 300 m of elevation difference between each elevation band. To account for natural heterogeneity within each elevation band, five separate sampling points were established at 20 m intervals at each elevation to capture the variation within the elevation range. This is a common practice in ecological studies to ensure robust and representative data [37,38,39,40,41]. To validate the consistency of samples within the same elevation, we conducted analyses with Bray–Curtis non-metric multidimensional scaling and clustering, both of which confirmed the consistency of samples within each elevation band (Appendix A). During the sampling process, all sampling points were located at least three meters away from the nearest tree. For each individual sample, five soil cores (0–10 cm depth, just below the litter layer) were collected from the corners and center of a 1 m × 1 m quadrat and then combined to form a composite sample. Visible roots and litter were removed from each fresh soil sample, which was then sieved through a 2 mm mesh. Each composite soil sample was subsequently divided into two subsamples: one stored at 4 °C for subsequent measurement of soil physical and chemical properties, and the other stored at −20 °C for soil environmental DNA extraction. In total, 90 soil samples were collected, with 30 samples from each mountain (6 elevation gradients × 5 replicates = 30 samples per mountain).

### 2.2. Environmental Data Sources and Measurement

Sixteen environmental attributes were measured at each sampling point (Appendix A). Climatic variables, including MAT and MAP, were obtained from the National Digital Climate Map compiled by the National Meteorological Center of the Korea Meteorological Administration [42]. Soil characteristics were measured at the National Instrumentation Center for Environmental Management, Seoul National University, following the standard protocols of the Soil Science Society of America. The measured soil properties included total organic carbon (TOC), total nitrogen (TN), NH_4_^+^, NO_3_^−^, total phosphorus (TP), available phosphorus (P_2_O_5_), pH value, moisture content, and texture, which were determined based on the proportions of sand, silt, and clay (Appendix A).

### 2.3. High-Throughput Sequencing and Amplicon Data Analysis

Environmental DNA was extracted from 0.25 g of soil, using a PowerSoil DNA extraction kit (Mo Bio Laboratories, Carlsbad, CA, USA), following the manufacturer’s instructions. High-throughput sequencing of the fungal ITS2 region was performed on the Illumina MiSeq platform (Illumina, Inc., San Diego, CA, USA) at the Centre for Comparative Genomics and Evolutionary Bioinformatics at Dalhousie University. The primer combination used was ITS86F (5′-GTGAATCATCGAATCTTTGAA-3′) and ITS4(R) (5′-TCC TCCGCTTATTGATATGC-3′) [43]. The polymerase chain-reaction conditions included an initial denaturation at 95 °C for 10 min, followed by 30 cycles at 95 °C for 30 s, 55 °C for 30 s, and 72 °C for 30 s, with a final extension at 72 °C for 7 min.

Raw ITS reads were obtained in the fastq format. Sequence data were processed using mothur (version 1.48.0, http://www.mothur.org; accessed on 27 October 2023), following the MiSeq SOP [44]. Forward and reverse reads were combined using the “make.contigs” command. Sequences shorter than 200 bp, chimeric sequences, and rare sequences were excluded from downstream analysis. High-quality sequences were assigned to operational taxonomic units (OTUs) at ≥97% similarity. This threshold has been widely used in the literature and provides a practical compromise for capturing fungal diversity without over-splitting OTUs [45,46,47]. Meanwhile, singletons and doubletons were removed to prevent the inclusion of potentially erroneous reads generated by sequencing errors. A smaller subset of 2764 high-quality sequences from the original set was randomly selected using the “sub.sample” command to ensure consistent sequencing depth across samples. The classification of each OTU was performed using the “classify.seqs” command with 1000 iterations against the UNITE [48] database, with a naive Bayesian bootstrap cutoff of 80% (Appendix A).

### 2.4. Statistical Analysis

Fungal functional guilds were assigned using FUNGuild (https://github.com/brendanf/FUNGuildR; accessed on 15 November 2023), which taxonomically parses fungal OTUs by analyzing the ecological guilds of sequencing databases [49]. Three broad trophic modes—pathotrophs, symbiotrophs, and saprotrophs—were defined based on the primary feeding habits of the fungi. Only “highly likely” guild assignments were considered. Otherwise, OTUs remained unclassified and categorized as “Others” (Appendix A). To avoid confusion arising from the terminology used in the FUNGuild database and to better convey the ecological significance of our results, we adopt the terms “mycorrhizal” and “parasitic” instead of “symbiotic” and “pathogenic”, respectively, except when directly referring to FUNGuild outputs.

Diversity indices, including the Shannon index, Chao index, Simpson index (1-D), and OTU richness, were estimated at different taxonomic levels and in different functional guilds. These indices were calculated using the diversity function of the “vegan” package in R (version 4.1.3) [50]. The Shannon index was used for linear regression analysis of diversity in response to environmental factors (Appendix A). The diversity of the fungal subcommunities was calculated using the same methods after rarefaction. To examine the influence of elevation on diversity, linear, quadratic, and cubic models were selected based on the Akaike information criterion (AIC). Spearman’s rank correlation was used to examine the relationships between diversity and environmental variables [51].

Variation partitioning analysis (VPA) was conducted using the “vegan” package in R to illustrate the independent or combined effects of three grouped environmental factors (climate, geography, and soil properties) on the variance in fungal alpha diversity (Shannon index) [52]. Environmental variables were categorized into three groups: climatic factors (MAT and MAP), geographical factors (elevation, longitude, and latitude), and soil factors (pH, TOC, TN, NH_4_^+^, NO_3_^−^, P_2_O_5_, moisture content, TP, sand, clay, and silt). To assess the relative importance of environmental variables in driving fungal diversity, analysis was performed using the “randomForest” package in R (version 4.7.1). Regression was performed using the “randomForest” function, and variable importance was determined based on the mean squared error increase (Inc MSE, %) value computed using the “importance” function.

## 3. Results

### 3.1. Alpha Diversity Patterns along Elevational Gradients

The biodiversity of the fungal community exhibited different patterns along the elevational gradients of the three mountains. On Mt. Halla, no significant patterns in alpha diversity were observed along the elevational gradient. This lack of significant variation extended across major fungal phyla and classes, as well as the three dominant functional guilds (Figure 2a; Appendix A). On Mt. Jiri, the fungal alpha diversity displayed a U-shaped distribution along the elevational gradient (R^2^ = 0.4556, *p* < 0.001). Specifically, Tremellomycetes and pathotrophs displayed this U-shaped pattern, reflecting the overall diversity trend. In contrast, Ascomycota and Dothideomycetes demonstrated a more complex pattern, with alpha diversity initially decreasing, then increasing, and finally decreasing (Figure 2b; Appendix A). On Mt. Seorak, the fungal alpha diversity showed a complex pattern characterized by an initial increase, followed by a decrease and then an increase along the elevational gradient (R^2^ = 0.2758, *p* < 0.05). Specifically, Ascomycota showed a linear decreasing trend in alpha diversity (*p* < 0.05), whereas the Zygomycota_class_Incertae_sedis diversity mirrored the overall trend with an initial increase, subsequent decrease, and final increase. However, no significant elevational gradient patterns were observed for the three most dominant functional guilds (Figure 2c; Appendix A).

### 3.2. Impact of Environmental Factors on Elevational Patterns of Fungal Diversity

The VPA results revealed that geographical factors, climatic factors, and soil properties collectively explained a substantial proportion of the variation in fungal community diversity. On Mt. Seorak, these factors accounted for the highest variance, explaining 67.1% of the variation in fungal community diversity. On Mt. Jiri and Mt. Halla, these factors explained 52.7% and 38% of the variation in diversity, respectively (Figure 3). Climatic factors were the primary explanatory variables for the variation in fungal community diversity on Mt. Jiri, with a variance explanation rate of 49.0%. Notably, 28.3% of the explanatory power was solely attributed to climatic factors, whereas the remaining portion (20.7%) was due to overlap between climatic factors, geographical factors, and soil properties. In contrast, the influence of climatic factors on fungal community diversity variation was lower on Mt. Halla (4.3%) and higher on Mt. Seorak (28.9%). Soil properties exhibited the highest explanatory power for fungal community diversity variation on Mt. Seorak, explaining 64.9% of the variance. Notably, 44.8% of the explanatory power was shared by geographical and climatic factors. Soil properties contributed to 31.1% and 17.1% of the variation in fungal community diversity on Mt. Halla and Mt. Jiri, respectively. In comparison to climatic and soil properties, geographical factors showed the lowest explanatory power for fungal community diversity variation on Mt. Jiri and Mt. Seorak, accounting for 24.4% and 20.3%, respectively. On Mt. Halla, geographical factors explained 11.8% of the variation.

Correlation and Random Forest variable importance analyses indicated that no common soil properties exhibited a strong correlation with fungal alpha diversity (represented by the Shannon index) on the three mountains (Figure 4a; Appendix A). However, soil TP emerged as the primary environmental variable that simultaneously influenced the Shannon index on Mt. Jiri (R = 0.59, *p* < 0.001) and Mt. Seorak (R = 0.55, *p* < 0.01), and P_2_O_5_ also had similar positive correlations on both mountains (Mt. Jiri: R = 0.59, *p* < 0.001; Mt. Seorak: R = 0.55, *p* < 0.01). Moreover, pH (R = 0.46, *p* < 0.05) and TOC (R = −0.56, *p* < 0.01) exhibited significant correlations with the Shannon index only on Mt. Seorak, while TN showed a significant positive correlation only on Mt. Jiri (R = 0.52, *p* < 0.01). Regarding climatic factors, there was no evidence of a correlation between MAP and the Shannon index, whereas MAT had a negative correlation with the Shannon index only on Mt. Jiri (R = −0.37, *p* < 0.05). Geographical factors had low explanatory power for fungal community diversity on all three mountains (Figure 4a). The correlations between environmental variables and fungal diversity that were expressed by OTU richness and the Chao index were consistent with those expressed by the Shannon index (Figure 4a).

### 3.3. Environmental Drivers of Dominant Taxa and Functional Group Diversity along Elevational Gradients

Pathotrophs, saprotrophs, and symbiotrophs were the three main functional guilds identified in this study (Figure 4b; Appendix A). The alpha diversity of the pathotrophs exhibited significant positive correlations with TP and P_2_O_5_ on both Mt. Jiri and Mt. Seorak (*p* < 0.01). Additionally, soil pH showed significant positive correlations with the pathotroph Shannon index on Mt. Halla and Mt. Seroka (*p* < 0.05). On Mt. Seorak, pathotroph diversity was also strongly correlated with elevation (R = 0.37, *p* < 0.05), latitude (R = 0.36, *p* < 0.05), TOC (R = −0.64, *p* < 0.001), and soil texture (sand: R = 0.62, *p* < 0.001; clay: R = −0.52, *p* < 0.01; silt: R = −0.54, *p* < 0.01). The environmental factors influencing the saprotroph groups were similar to those affecting pathotrophs. Additionally, TP had a positive correlation with saprotroph diversity on both Mt. Jiri and Mt. Seorak (*p* < 0.05), whereas P_2_O_5_ showed a positive correlation with saprotroph diversity only on Mt. Jiri (R = 0.62, *p* < 0.001). On Mt. Seorak, pH (R = 0.38, *p* < 0.05), TOC (R = −0.55, *p* < 0.01), and soil texture (sand: R = 0.55, *p* < 0.01; silt: R = −0.52, *p* < 0.01) were significantly correlated with saprotroph diversity. Notably, TP and AP exhibited significant positive correlations with the diversity of all three functional groups on Mt. Jiri (*p* < 0.01). Unlike soil properties, climatic and geographic factors were not significantly correlated with fungal functional diversity.

We investigated the influence of environmental factors on the major fungal taxonomic groups (Figure 4c; Appendix A). On Mt. Jiri, the alpha diversity of Ascomycota, Basidiomycota, and Dothideomycetes was positively correlated with TP, P_2_O_5_, and TN (*p* < 0.05). Meanwhile, Leotiomycetes and Tremellomycetes showed significant correlations with climatic (MAT and MAP) and geographic variables (elevation, latitude, and longitude) (*p* < 0.01), but variables that positively influenced Leotiomycetes negatively affected Tremellomycetes and vice versa. On Mt. Seorak, the alpha diversity of Zygomycota and Zygomycota_class_Incertae_sedis was positively correlated with TP, P_2_O_5_, and pH (*p* < 0.05), whereas that of Ascomycota was significantly positively correlated with climatic and geographic variables (*p* < 0.01). Additionally, the alpha diversity of Zygomycota, Leotiomycetes, and Dothideomycetes was positively correlated with NO_3_^−^ (*p* < 0.05).

## 4. Discussion

Species diversity varies along elevational gradients, with fungal communities often exhibiting particularly complex patterns. This complexity stems primarily from the high sensitivity of fungi to changes in the soil microenvironment [21]. Fungal communities quickly respond to subtle changes in microclimatic conditions, soil chemistry, and biotic interactions, resulting in complex and variable diversity patterns along elevational gradients [53]. Our study, focusing on three mountains reaching approximately 2000 m above sea level at different latitudes on the Korean Peninsula, provides insights into the diverse patterns of fungal community diversity along elevational gradients. The three mountains exhibited distinctly different diversity trends with increasing elevation. As elevation increased, Mt. Seorak’s fungal alpha diversity first increased and then decreased, showing the highest species diversity at mid-elevation zones. Conversely, Mt. Jiri displayed a U-shaped distribution, with the lowest diversity at mid-elevation zones, while Mt. Halla showed no significant change in fungal alpha diversity with changing elevation. Classic ecological theories have posited that diversity generally decreases with elevation, suggesting that a similar trend would exist for fungi, with fungal abundance decreasing along mountain elevational gradients [21]. This phenomenon is typically associated with the increasing harshness of the environment at higher elevations [54]. For example, studies have observed a monotonic decline in soil fungal diversity along elevations ranging from 700 to 2600 m on Changbai Mountain [20]. However, as ecological research has advanced, more studies have found that peak diversity occurs at mid-elevation zones. For example, ectomycorrhizal fungi on Mt. Fuji in Japan exhibit a hump-shaped distribution along the elevational gradient [55,56]. Several theories have been proposed to explain the peak fungal diversity at mid-elevation zones. The “community overlap” hypothesis suggests that mid-elevation zones function as transitional areas between the distinct environments of the mountain top and base, resulting in higher species diversity due to the inclusive nature of transitional habitats. Another widely recognized explanation is the “mid-domain effect”, which posits that if species ranges are randomly distributed between the mountain top and base boundaries, the mid-elevation zone will exhibit the highest species overlap [57]. While these theories emphasize the importance of deterministic processes, such as climatic and geographical factors, in influencing soil fungal diversity, Wang et al. [13] suggested that stochastic processes can also substantially affect mountain fungal community diversity under certain conditions. Although a U-shaped pattern (with the lowest diversity at mid-elevation zones) is uncommon, studies on fungal diversity in Korean pine forests on Changbai Mountain have found the lowest community diversity at mid-elevation zones [58], which is consistent with our findings on Mt. Jiri. Additionally, the primary vegetation from low- to mid-elevation zones on Mt. Jiri is *Pinus densiflora* [34], indicating that soil properties influenced by Korean pines may negatively affect fungal diversity, warranting further research. Birch trees (*Betula* spp.) play a significant role in shaping soil properties and influencing fungal diversity. Birch trees can alter soil pH, organic matter content, and nutrient availability, thereby affecting fungal biomass and communities [59,60]. This impact is particularly important at mid to high elevations, where birch trees are more prevalent. On Mt. Halla, the fungal community alpha diversity did not show significant changes with elevation, differing from previously observed distribution patterns. However, a study in Southeastern Tibet reported a similar lack of a gradient pattern in fungal diversity, generally attributed to the variation in habitats along the elevational gradient [61].

Elevational gradients are associated with changes in various environmental factors. As elevation increases, many environmental variables, such as temperature, humidity, soil pH, and soil trace-element content, also change accordingly [62]. For example, although lower temperatures and higher humidity at high elevations can support specific fungal communities, reduced soil nutrient levels (such as TN and TOC) and extreme climatic conditions limit the survival of many fungal species [63]. Understanding the distribution and diversity of fungi at different elevations provides crucial insights into how mountain ecosystems respond to environmental changes [64]. Our findings confirmed the important role of climatic, soil, and geographical factors in shaping the elevational gradient of fungal community alpha diversity.

Climate is a crucial factor controlling fungal community diversity [65,66,67], and it indirectly controls the relative abundance of fungal community diversity through its strong influence on ecosystem types, vegetation, and soil properties [68]. In our study, fungal community alpha diversity showed a significant correlation with not only MAP but also MAT. Specifically, on Mt. Jiri, fungal community alpha diversity was negatively correlated with MAT, consistent with previous studies indicating that temperature directly affects fungal communities through physical tolerance and enzymatic processes and indirectly affects them through its impact on vegetation turnover [69,70,71,72]. Compared to climatic factors, soil properties have a more significant impact on soil fungal diversity. Soil nutrient availability has been shown to significantly influence fungal community composition [73,74,75]. In particular, P is an important energy source for microorganisms, significantly influencing the elevational distribution patterns of soil microbes by determining their metabolism [76,77,78]. Our study showed that TP and P_2_O_5_ positively affected the alpha diversity of fungal communities and different taxonomic groups across the three mountains, possibly due to the preference of certain fungi for specific elements [66]. Many studies have found that soil nutrients such as TP [79] and available N [80] affect fungal growth [73,81,82] and may limit the maximum potential fungal diversity in the soil [80]. Additionally, phosphates are crucial for fungal proliferation, stress response, cell wall synthesis, and carbon metabolism, serving as key mediators in interactions with other organisms. The availability of phosphates influences fungal adaptation to environmental stress and host interactions, directly impacting the structure and diversity of soil fungal communities. Furthermore, phosphates act as signaling molecules within fungal cells, regulating nutrient acquisition, stress responses, and metabolic processes, enabling fungi to adjust their diversity and distribution in various soil environments [83]. In our study, TN and TOC also influenced the alpha diversity of fungal communities on different mountains. In contrast, soil pH did not significantly affect fungal biodiversity, likely because of fungi’s broad optimal pH range for growth [84]. The influence of geographical factors on soil microbial communities depends on the spatial scale [85,86,87]. Our study focused on the elevational gradients of individual mountains in a local-scale investigation. Therefore, the geographical factors in this study were insufficient to affect the fungal community diversity more than the environmental factors. This finding aligns with previous research indicating that, at regional scales, geographical factors explain some variations in soil fungal diversity [88], whereas at local scales, fungal diversity changes are primarily regulated by environmental factors [89].

It is well understood that underground microbial communities are largely determined by the diversity and composition of the aboveground plant species [88,90]. For example, Tedersoo et al. [88] found that, as the proportion of *Pinus sylvestris* increases in forests, tree diversity has a negative effect on fungal diversity, with species-identity effects dominating aboveground–belowground relationships beyond plant diversity. Therefore, the influence of vegetation on fungal communities cannot be ignored. In our study, we sampled the bulk soil rather than rhizosphere soil on the three mountains to minimize the influence from vegetation on fungi communities, making it possible to explore the effects of a more diverse series of environmental factors. Indeed, the relative low abundance of functional groups of arbuscular mycorrhizal fungi and Ectomycorrhizal fungi confirmed the success of our sampling plan.

Although the three main functional groups (i.e., pathotrophs, saprotrophs, and symbiotrophs) exhibited varied responses to environmental constraints, the saprotrophs responded to environmental factors in a manner similar to that of pathotrophs on all the three mountains. Future studies should be conducted to comprehensively understand their effects on fungal diversity. The diversity of parasitic fungi on Mt. Jiri exhibited a U-shaped distribution, consistent with the findings of previous research. TP was significantly positively correlated with the diversity of parasitic fungi. This may be because P, as an essential nutrient, promotes microbial growth and reproduction. Parasitic fungi obtain nutrients by damaging host cells; thus, they thrive and reproduce easily in high-P environments [83]. On Mt. Seorak, the diversity of parasitic fungi was also found to be influenced by pH, TOC, and soil texture (sand, clay, and silt). This may be because parasitic fungi tend to thrive in nutrient-rich environments, where these soil properties provide favorable conditions for their growth [49,91]. Notably, the diversity patterns of the three functional groups on Mt. Jiri and Mt. Seorak were consistent with the overall trend and were significantly correlated with TP, indicating the important role of fungal functional groups in determining the overall diversity. Interestingly, our study reveals that, within each mountain, the environmental factors influencing the diversity patterns of different fungal functional groups are generally consistent. However, significant differences exist in the environmental factors affecting the diversity of these fungal functional groups across the three mountains. This variability may be attributed to the distinct fungal community diversity patterns observed in each mountain.

## 5. Conclusions

This study investigated the patterns of soil fungal community diversity along elevational gradients on three major mountains in the Korean Peninsula: Mt. Halla, Mt. Jiri, and Mt. Seorak. By employing high-throughput sequencing to analyze fungal communities, we elucidated the diversity patterns at different elevations and identified the key environmental factors influencing these patterns. The results revealed distinct elevational diversity patterns of fungal communities on each mountain. Geographic distance, climatic factors, and soil properties collectively explained a significant portion of the variation in fungal community diversity. Among these factors, soil properties, particularly total phosphorus and available phosphorus, emerged as critical determinants of fungal diversity, highlighting the importance of soil nutrients in shaping fungal communities. These findings offer new insights into the biogeographical distribution of fungi and the ecological processes driving fungal diversity in temperate forest ecosystems. Overall, our study enhances the understanding of fungal diversity dynamics along elevational gradients and underscores the necessity of considering multiple environmental factors when assessing microbial biodiversity. The results provide valuable references for future ecological research and conservation efforts of similar biotic communities, especially in the context of changing environmental conditions.

## Figures and Tables

**Figure 1 jof-10-00556-f001:**
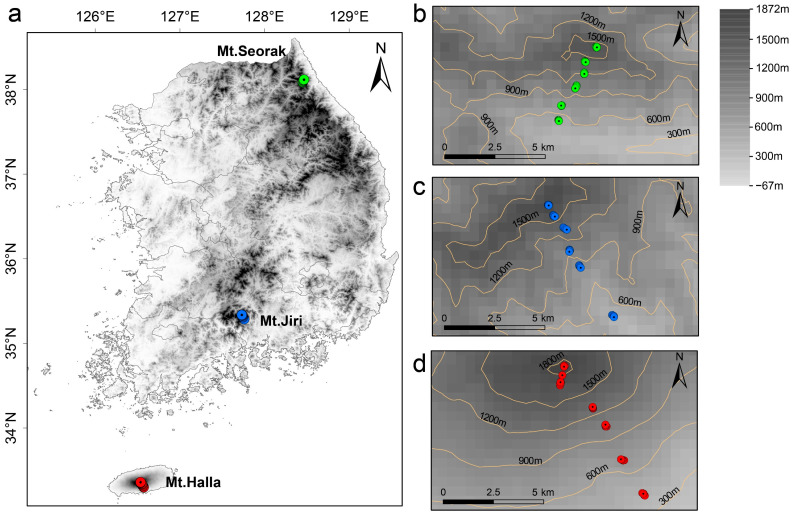
Sampling site (**a**) and schematic diagram of sampling points: Mt. Seorak (**b**), Mt. Jiri (**c**), and Mt. Halla (**d**). The green, blue, and red dots represent Mt. Halla, Mt. Jiri, and Mt. Seorak, respectively.

**Figure 2 jof-10-00556-f002:**
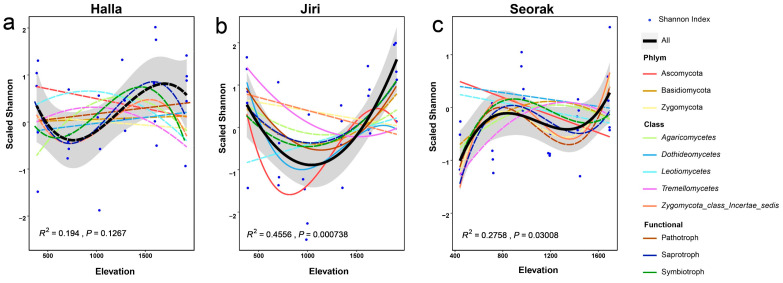
Elevational patterns of fungal community diversity, including overall and specific phylum, class, and functional diversity, indicated by blue dots, black fitted curves, and gray confidence intervals, respectively. Panels (**a**–**c**) represent alpha diversity (Shannon index transformed by z-score) for Mt. Halla, Mt. Jiri, and Mt. Seorak. The quadratic model was selected based on the AIC (refer to Appendix A). Blue dots represent the values of the Shannon index transformed by z-score at each sampled sites for each mountain. Solid lines indicate significant trends, while dashed lines represent non-significant trends.

**Figure 3 jof-10-00556-f003:**
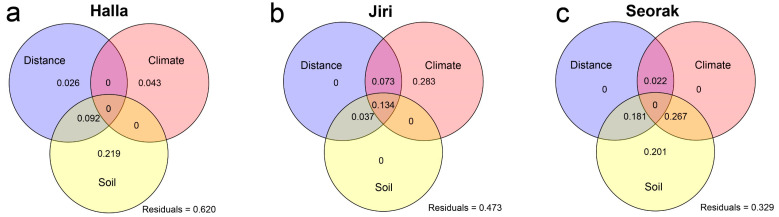
Variance partitioning analysis (VPA) demonstrates the combined influence of geographical, climatic, and soil properties on the fungal community alpha diversity indices across the three mountains. Panels (**a**–**c**) represent Mt. Halla, Mt. Jiri, and Mt. Seorak, respectively. Geographical factors include longitude and latitude; climatic factors include mean annual temperature (MAT) and mean annual precipitation (MAP); and soil properties include pH, total organic carbon (TOC), total nitrogen (TN), NH_4_^+^, NO_3_^−^, available phosphorus (P_2_O_5_), moisture content, total phosphorus (TP), and soil texture (sand, silt, and clay).

**Figure 4 jof-10-00556-f004:**
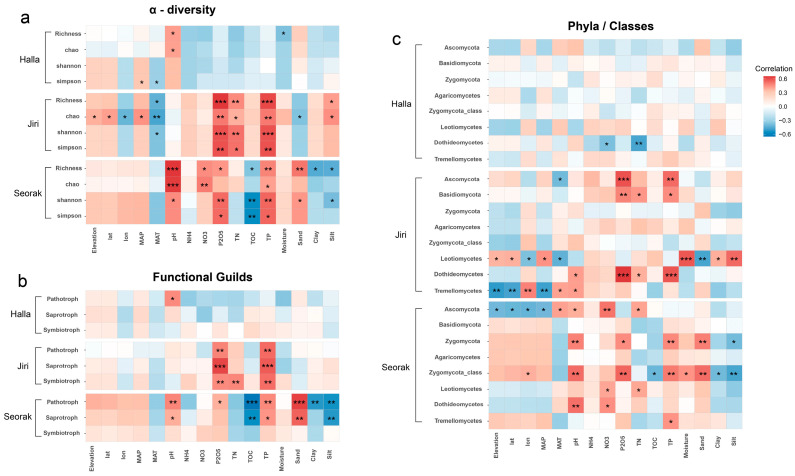
The correlation between fungal alpha diversity (**a**), including richness and Chao, Shannon, and Simpson indices), fungal functions (**b**), and the alpha diversity of dominant fungal phyla and genera (**c**) with environmental factors across the three mountains. * *p* < 0.05, ** *p* < 0.01, and *** *p* < 0.001.

## Data Availability

The original contributions presented in the study are included in the article/Appendix A, further inquiries can be directed to the corresponding authors.

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
