# Peer review of "Elevational Variation in and Environmental Determinants of Fungal Diversity in Forest Ecosystems of Korean Peninsula"

_jof, 2024, doi:10.3390/jof10080556_

Round 1

Reviewer 1 Report

The question, the study design, and the data (for the most part) are a fine piece of work. However, the authors need to do a better job at tackling the data and presenting an analysis that is more meaningful. Major questions to address: 1. Much more attention must be given to the within-elevation level of variation — the replicate sampling points. Without drawing the reader's attention to this, no other levels of analysis (elevation, mountain) are properly addressed. The only place where this variation level is visible (though not explained appropriately) are there the blue dots in Figure 2, and one can see how different the replicates are, within any elevation! It would be important to compute a similarity index (Bray-Curtis for example) over all samples, or at least for each mountain, and display the results in nMDS graph. Also the VPA should state explicitly what is the component that is attributed to community variation between replicate sampling points: it might correspond to the Residuals components in Figure 3, but the reader is left to guessing. The results text must explain this, and also explain which source of variation the authors interpret in this figure: is it elevation? 2. The OTU table (Table S2) features 5,085 OTUs, of which some 1,000 are as good as noise (see Detail comments, below). It is suggested that the authors run their analyses with a revised OTU table. 3. Outdated use of the words symbiotic and pathogenic. The first can be parasitic as well (and including also, by the way, endophytes, mixed/alternating biotrophic modes...), and the second is context-dependent (for example, natural versus managed forests). Use of the ecologically comparable terms mycorrhizal and parasitic, respectively, is strongly advised. The wording of the FUNGuild database is likewise inadequate, and while one can use theirs in the context of the FUNGuild outputs, elsewhere this confusion is inacceptable. Please check the authoritative web site https://mycorrhizas.info/#symb and correct accordingly (some indications are suggested in the PDF file as red text on the margin). 4. The authors provide relevant information about the vegetation, but ignore this all-important factor in the analysis of the results, for example in the VPA and Random Forest correlation study. Personally, I feel uncomfortable with the statements (lines 53-55 for 'microorganisms' and lines 298-300 for 'fungi'), that the diversity patterns are more complex than in plants and animals. These are vague statements, in spite of the citations in presumed support (which is none, actually, I checked). The same lack of support for the statement in line 395-6. By the way, reference 80 identifies K, not P. Using a simple rule for OTUs that are "too rare" (up to 5 reads total, majority of singletons), I located 1,052 that may represent noise, which is an all-too-large proportion. I enclose in the attached zip file the OTU table with flags showing the extent of such suspect OTUs. The authors might do well to work out a rule (not necessarily the one I have used) that enables them to have a "cleaner" OTU table. Line 118: the 3.05 ºC MAT for Mt. Seorak is at what altitude? In this section, the MATs given should be at comparable altitudes. Line 130: I like the 5-fold replications, but there is no meaningful analysis of the degree of variation between them in each mountain x elevation combination. See the Major comments for more ideas on this. Line 164: the >97% similarity is notably unsuited to fungal ITS regions. However, this can be considered a "matter of taste". Line 171: What is the proportion of OTUs that were excluded from analysis because of uncertainties in the FUNGuild assignment? Are these the 'others' in figure S2, or those who belong in two or more categories? These 'others' must be clarified. Figure 2 caption: the blue dots (Shannon index for each replicate sampling point) must be described. Figure 3: the empty spaces in figure 3 should have a 0. Lines 235-237: the statement is false, because 24.4% in Jiri is not the lowest component, contrary to what the text says! (soil accounts for 17.1% in Jiri) A statement in lines 250-251 reiterates this idea and should be removed because it is misplaced. Lines 251, 253: Figure 4a and not Figure 3a. Line 361: I find it somewhat hard to understand how P affects the 'elevational distribution patterns', unless it had a parallel variation with altitude. Or perhaps this phrase is less clear than it needs to be. Checking Table S1 did not show such a relationship of P with elevation. On the PDF I suggest its removal, but perhaps a better phrasing clears out the problem. Figure 4a and lines 251-253: the Simpson's D value is actually the opposite of species diversity. Usually it is 1-D that is used to check for a comparison with other alpha-diversity indices. It is very likely that the authors are reporting based on D instead of 1-D, and that is why one sees the Simpson colouring inversely correlated with the other three in each mountain. Figure 4b: I disagree completely with the reading made by the authors on the results. What is really consistent is the difference between the mountains, while, in each mountain, the trophic modes are roughly similar in their correlation with environmental variables. Lines 383-4: no evidence for this, since many (possibly most) rhizosphere mycelia extend beyond this narrow layer. Lines 406-410: the explanations are conceptually flawed, because adaptability should be considered at species level (as part of each species niche), not at the phyllum level.

Author Response

We express our profound gratitude to the reviewer for dedicating their valuable time to review our manuscript. We have made many corrections and edits throughout the manuscript. To facilitate an efficient review process, we have included all the comments within this response letter, ensuring ease of reference and comprehensive understanding. We also would like to express our sincere gratitude for you highlighting numerous minor corrections in our original manuscript to enhance the quality of the article. We have carefully reviewed and addressed each of your suggested corrections in the revised manuscript.

Reviewer(s)' Comments to Author:

The question, the study design, and the data (for the most part) are a fine piece of work. However, the authors need to do a better job at tackling the data and presenting an analysis that is more meaningful.

The major comments:

1-1. Much more attention must be given to the within-elevation level of variation — the replicate sampling points. Without drawing the reader's attention to this, no other levels of analysis (elevation, mountain) are properly addressed. The only place where this variation level is visible (though not explained appropriately) are there the blue dots in Figure 2, and one can see how different the replicates are, within any elevation!

We appreciate your attention to the details of our sampling methodology and data representation. In this study, we employed a rigorous sampling strategy to capture the within-elevation variation by collecting five samples at each elevation level. This approach was designed to account for the natural heterogeneity within each elevation band, which is a common practice in ecological studies to ensure robust and representative data[1-5]. While the within-elevation variation is evident in Figure 2, the primary focus of our study was on the broader elevational and inter-mountain patterns of fungal diversity. The variation among the samples from the same elevations reflects our effort to account for the inherent heterogeneity and to minimize the limitations of representing species diversity at the same elevation. This provides valuable insights into the natural variability within each elevation level. We have now revised the methods part of our manuscript to help readers understand the full scope of our analysis and the robustness of our findings. (L133-L140)

1-2. It would be important to compute a similarity index (Bray-Curtis for example) over all samples, or at least for each mountain, and display the results in nMDS graph.

We greatly appreciate your input on enhancing the depth of our analysis. While the Bray-Curtis similarity index is excellent for analyzing species composition and community structure across samples, and nMDS is helpful for visualizing these differences, they are not directly aligned with our primary research objective. The primary focus of our study is to investigate the diversity trends of fungal communities along elevational gradients, rather than to compare community composition between individual samples. Our current approach emphasizes the analysis of fungal diversity patterns using alpha diversity indices, which directly reflect the richness and evenness of fungal communities at different elevations. These indices provide a clear understanding of how fungal diversity varies with elevation and among different mountains. To address your suggestion, we have included the OTU table as supplementary material (Supplementary Table S2) to provide a detailed view of the fungal community structure. We hope by doing so it will allow interested readers to explore community composition differences while maintaining the focus of the main text on diversity trends.

1-3. Also the VPA should state explicitly what is the component that is attributed to community variation between replicate sampling points: it might correspond to the Residuals components in Figure 3, but the reader is left to guessing. The results text must explain this, and also explain which source of variation the authors interpret in this figure: is it elevation?

We measured a series of environmental variables and categorized them into three groups: climatic factors (MAT and MAP), geographical factors (elevation, longitude, and latitude), and soil properties (pH, TOC, TN, NH₄⁺, NO₃⁻, P₂O₅, moisture content, TP, sand, clay, and silt). We then conducted VPA to illustrate the independent or combined effects of these three grouped environmental factors (climate, geography, and soil properties) on the variance in fungal alpha diversity (Shannon index). Additionally, we performed random forest analysis to assess the relative importance of these environmental variables in driving fungal diversity. We believe that by combining the results of these two analyses, readers will gain a comprehensive understanding of which environmental factors play a significant role in driving fungal diversity in this study (L201-L211).

2. The OTU table (Table S2) features 5,085 OTUs, of which some 1,000 are as good as noise (see Detail comments, below). It is suggested that the authors run their analyses with a revised OTU table.

We understand the importance of ensuring the quality and reliability of our OTU data. To address your concern, we would like to clarify that our data processing has already excluded singletons and doubletons to minimize the inclusion of potentially spurious sequences that could be considered noise.

Although there are still low-abundance OTUs remaining in our dataset, these OTUs are considered rare communities. Their low abundance is due to our data standardization process using the subsample command in mothur, which involves randomly selecting a subset from the original dataset. This approach ensures consistent sequencing depth across samples, thereby eliminating biases caused by differences in sequencing effort.

While low in abundance, rare communities provide valuable ecological information and insights into the diversity and composition of fungal communities. They contribute to the overall diversity and ecological complexity of the fungal communities. Removing these rare OTUs could lead to a loss of significant data, potentially affecting the interpretation of biodiversity patterns[6]. To ensure clarity for the readers, we have revised this part in the Methods section to explicitly state our approach regarding singletons and doubletons. (L174-L179)

3. Outdated use of the words symbiotic and pathogenic. The first can be parasitic as well (and including also, by the way, endophytes, mixed/alternating biotrophic modes...), and the second is context-dependent (for example, natural versus managed forests). Use of the ecologically comparable terms mycorrhizal and parasitic, respectively, is strongly advised. The wording of the FUNGuild database is likewise inadequate, and while one can use theirs in the context of the FUNGuild outputs, elsewhere this confusion is inacceptable. Please check the authoritative web site https://mycorrhizas.info/#symb and correct accordingly (some indications are suggested in the PDF file as red text on the margin).

Thank you very much for your valuable suggestions. To avoid confusion arising from the terminology used in the FUNGuild database and to better convey the ecological significance of our results, we have revised the manuscript to use the terms "mycorrhizal" and "parasitic" instead of "symbiotic" and "pathogenic," respectively, except when directly referring to FUNGuild outputs. We have revised the text accordingly in the manuscript. (L69-L71, L187-191, L424-L431)

4. The authors provide relevant information about the vegetation. But ignore this all-important factor in the analysis of the results, for example in the VPA and Random Forest correlation study.

In this study, we sampled bulk soil rather than rhizosphere soil, thereby minimizing the direct influence of vegetation on fungal communities. By sampling bulk soil, we aimed to capture the overall soil fungal community that is influenced by a variety of environmental factors, not just vegetation. While this approach may not completely exclude the effects of vegetation, it reduces the direct impact of root-associated fungal communities, providing a more generalized view of fungal diversity patterns. This method is considered applicable for exploring a broader range of environmental factors beyond vegetation[5, 7, 8]. We fully recognize the significant impact that aboveground communities, particularly plants, have on belowground communities. Indeed, we plan to further our research in the future by elucidating the effects of vegetation diversity and structure on the belowground communities in mountain forests on the Korean Peninsula. We believe this will provide additional insights into the complex interactions between plants and soil fungi. In the revised manuscript, we have now included a brief discussion to remind the readers not to overlook this important factor in the analysis of the results. For example, the effects of vegetation might correspond to the residual components in the VPA. (L410-L420)

Detail comments

1. Personally, I feel uncomfortable with the statements (lines 53-55 for 'microorganisms' and lines 298-300 for 'fungi'), that the diversity patterns are more complex than in plants and animals. These are vague statements, in spite of the citations in presumed support (which is none, actually, I checked).

We appreciate your concerns and recognize that these statements may have been perceived as vague and we appreciate the opportunity to clarify them. We have revised these parts in the revised manuscript. (L58-L60; L318-L319)

2. reference 80 identifies K, not P. The same lack of support for the statement in line 395-6.

Thank you for your careful review and for pointing out the inaccuracies in our citations. We have now replaced the incorrect reference and revised this part. (L390-L397, L424-L432)

3. Using a simple rule for OTUs that are "too rare" (up to 5 reads total, majority of singletons), I located 1,052 that may represent noise, which is an all-too-large proportion. I enclose in the attached zip file the OTU table with flags showing the extent of such suspect OTUs. The authors might do well to work out a rule (not necessarily the one I have used) that enables them to have a "cleaner" OTU table.

Thank you for your comment. We have addressed similar concerns in our response to your second major comment (Comment 2). For your convenience, we would like to refer you to our previous detailed response on this matter.

4. Line 118: the 3.05 ºC MAT for Mt. Seorak is at what altitude? In this section, the MATs given should be at comparable altitudes.

Thank you for your insightful comment. We have now revised the text to clarify that the 3.05 ºC MAT is an average temperature across the entire elevational range and to direct readers to the supplementary table for detailed MAT data at specific altitudes (L123-L124, L150, Supplementary Table S1).

5. Line 130: I like the 5-fold replications, but there is no meaningful analysis of the degree of variation between them in each mountain x elevation combination. See the Major comments for more ideas on this.

Thank you for your positive feedback regarding the 5-fold replications. We have addressed this concern in detail in our response to your first major comment (Comment 1-1). For your convenience, please refer to our previous detailed response on this matter.

6. Line 164: the >97% similarity is notably unsuited to fungal ITS regions. However, this can be considered a "matter of taste".

We appreciate your comment on the similarity threshold for fungal ITS regions. We understand that the choice of similarity threshold can be a topic of debate and may vary depending on the specific goals of the study. In our study, we chose a >97% similarity threshold for clustering OTUs based on standard practices in fungal ITS region analysis, which allows for a balance between specificity and inclusivity. This threshold has been widely used and provides a practical compromise for capturing fungal diversity without over-splitting OTUs. We acknowledge that different thresholds (e.g., 99% or higher) can be used to achieve higher resolution, but these may also lead to an increased number of OTUs and potential challenges in downstream analyses[9]. Our choice of >97% similarity aimed to provide a comprehensive yet manageable dataset for analyzing fungal community diversity across the sampled elevations[10-12]. We have now made further explanation on this issue in the revied manuscript. (L174-L179)

7. Line 171: What is the proportion of OTUs that were excluded from analysis because of uncertainties in the FUNGuild assignment? Are these the 'others' in figure S2, or those who belong in two or more categories? These 'others' must be clarified.

We appreciate your concern regarding the clarity of OTU categorization. The 'others' in Figure S2 represents OTUs that could not be assigned to a specific guild with high confidence. To enhance clarity for the readers, we have revised the relevant sections in the manuscript. (L187)

8. Figure 2 caption: the blue dots (Shannon index for each replicate sampling point) must be described.

We have now revised the legend in the updated manuscript (L236-L237)

9. Figure 3: the empty spaces in figure 3 should have a 0.

We have now filled in the spaces without values in Figure 3 with zeros (Figure 3).

10. Lines 235-237: the statement is false, because 24.4% in Jiri is not the lowest component, contrary to what the text says! (soil accounts for 17.1% in Jiri) A statement in lines 250-251 reiterates this idea and should be removed because it is misplaced.

Thank you for pointing out the inaccuracies in our statements. We have revised the text accordingly to accurately reflect the data. (Figure 3, L254-L258)

11. Lines 251, 253: Figure 4a and not Figure 3a.

We have now corrected these in the revised manuscript (L271, L273).

12. Line 361: I find it somewhat hard to understand how P affects the 'elevational distribution patterns', unless it had a parallel variation with altitude. Or perhaps this phrase is less clear than it needs to be. Checking Table S1 did not show such a relationship of P with elevation. On the PDF I suggest its removal, but perhaps a better phrasing clears out the problem.

We appreciate your comment. In this study, we observed that phosphorus levels significantly correlate with overall fungal diversity indices, indicating that phosphorus is an important factor influencing fungal diversity (Figure. 4a; Table S6), even though it does not vary systematically with elevation. According to Bhalla et al. (2022), phosphate is crucial for fungal proliferation, stress responses, cell wall synthesis, and carbon metabolism, serving as a key mediator in interactions with other organisms. Its regulation affects fungal growth, diversity, and functionality across various environments. Phosphate availability influences fungal adaptation to environmental stress and host interactions, directly impacting the structure and diversity of soil fungal communities. Additionally, phosphate acts as a signaling molecule in fungal cells, regulating nutrient acquisition, stress responses, and metabolic processes, thereby enabling fungi to adjust their diversity and distribution in different soil environments. We have now revised our discussion section accordingly in the revised manuscript. (L390-L397)

13. Figure 4a and lines 251-253: the Simpson's D value is actually the opposite of species diversity. Usually it is 1-D that is used to check for a comparison with other alpha-diversity indices. It is very likely that the authors are reporting based on D instead of 1-D, and that is why one sees the Simpson colouring inversely correlated with the other three in each mountain.

We appreciate your careful review and for pointing out this oversight. We acknowledge that we incorrectly showed the results based on D instead of 1-D. In our revised analysis, we have corrected this error and now use 1-D for the Simpson index. We have made the necessary corrections throughout our revised manuscript. (L192, Figure 4, Supplementary Table S3 & S6).

14. Figure 4b: I disagree completely with the reading made by the authors on the results. What is really consistent is the difference between the mountains, while, in each mountain, the trophic modes are roughly similar in their correlation with environmental variables.

We appreciate your perspective on our results and understand the importance of accurately interpreting the data. We agree that while the correlations between the three trophic modes and environmental variables are generally similar within each mountain, there are significant differences in their relationships across different mountains. We have revised the manuscript to better emphasize these points. (L435-L440)

15. Lines 383-4: no evidence for this, since many (possibly most) rhizosphere mycelia extend beyond this narrow layer.

Thank you for raising this important consideration. We recognize the significant impact that aboveground communities, particularly plants, have on belowground communities. Therefore, we sampled bulk soil rather than rhizosphere soil in this study, thereby minimizing the direct influence of vegetation on fungal communities. By sampling bulk soil, we aimed to capture the overall soil fungal community that is influenced by a variety of environmental factors, not just vegetation. While this approach may not completely exclude the effects of vegetation, it is considered applicable for exploring a broader range of environmental factors beyond vegetation [13, 14]. In the revised manuscript, we have included a brief discussion to remind readers not to overlook the impact of vegetation may hold in this study. (L410-L420)

16. Lines 406-410: the explanations are conceptually flawed, because adaptability should be considered at species level (as part of each species niche), not at the phylum level.

We appreciate your feedback and understand the importance of considering adaptability at the species level, as it is a crucial aspect of each species' ecological niche. However, we believe that discussing adaptability at higher taxonomic levels, such as the phylum level, can also provide valuable insights, depending on the scientific questions being addressed. In our study, we aimed to identify general trends and patterns in fungal diversity and distribution across different elevational gradients. Examining adaptability at the phylum level allows us to capture broader ecological responses and adaptive strategies that may be shared among related species within a phylum. These broader patterns can help us understand how major fungal groups respond to environmental gradients and how these responses contribute to overall ecosystem functioning.

  1. Dickie, I. A., S. Boyer, H. L. Buckley, R. P. Duncan, P. P. Gardner, I. D. Hogg, R. J. Holdaway, G. Lear, A. Makiola, S. E. Morales, J. R. Powell, and L. Weaver. "Towards Robust and Repeatable Sampling Methods in Edna-Based Studies." Mol Ecol Resour (2018).
  2. Rong, She, Qi Fu-Liang, Chen Yi-Ting, Zhou Fa-Ping, Deng Wei, Lu Ya-Xian, Huang Zhi-Pang, Yang Xiao-Yan, and Xiao Wen. "Soil Sampling Methods for Microbial Study in Montane Regions." Global Ecology and Conservation 47 (2023).
  3. Zou, Shuqi, Jonathan Adams, Zhi Yu, Nan Li, Dorsaf Kerfahi, Binu Tripathi, Changbae Lee, Teng Yang, Itumeleng Moroenyane, Xing Chen, Jinsoo Kim, Hyun Jeong Kwak, Matthew Chidozie Ogwu, Sang-Seob Lee, and Ke Dong. "Stochasticity Dominates Assembly Processes of Soil Nematode Metacommunities on Three Asian Mountains." Pedosphere 33, no. 2 (2023): 331-42.
  4. Yu, Z., S. Zou, N. Li, D. Kerfahi, C. Lee, J. Adams, H. J. Kwak, J. Kim, S. S. Lee, and K. Dong. "Elevation-Related Climatic Factors Dominate Soil Free-Living Nematode Communities and Their Co-Occurrence Patterns on Mt. Halla, South Korea." Ecol Evol 11, no. 24 (2021): 18540-51.
  5. Yu, Zhi, Changbae Lee, Dorsaf Kerfahi, Nan Li, Naomichi Yamamoto, Teng Yang, Haein Lee, Guangyin Zhen, Yenan Song, Lingling Shi, and Ke Dong. "Elevational Dynamics in Soil Microbial Co-Occurrence: Disentangling Biotic and Abiotic Influences on Bacterial and Fungal Networks on Mt. Seorak." Soil Ecology Letters 6, no. 4 (2024).
  6. Zhang, Z., Y. Lu, G. Wei, and S. Jiao. "Rare Species-Driven Diversity-Ecosystem Multifunctionality Relationships Are Promoted by Stochastic Community Assembly." mBio 13, no. 3 (2022): e0044922.
  7. Shen, C., A. Gunina, Y. Luo, J. Wang, J. Z. He, Y. Kuzyakov, A. Hemp, A. T. Classen, and Y. Ge. "Contrasting Patterns and Drivers of Soil Bacterial and Fungal Diversity across a Mountain Gradient." Environ Microbiol 22, no. 8 (2020): 3287-301.
  8. Siles, J. A., and R. Margesin. "Abundance and Diversity of Bacterial, Archaeal, and Fungal Communities Along an Altitudinal Gradient in Alpine Forest Soils: What Are the Driving Factors?" Microb Ecol 72, no. 1 (2016): 207-20.
  9. Koljalg, U., R. H. Nilsson, K. Abarenkov, L. Tedersoo, A. F. Taylor, M. Bahram, S. T. Bates, T. D. Bruns, J. Bengtsson-Palme, T. M. Callaghan, B. Douglas, T. Drenkhan, U. Eberhardt, M. Duenas, T. Grebenc, G. W. Griffith, M. Hartmann, P. M. Kirk, P. Kohout, E. Larsson, B. D. Lindahl, R. Lucking, M. P. Martin, P. B. Matheny, N. H. Nguyen, T. Niskanen, J. Oja, K. G. Peay, U. Peintner, M. Peterson, K. Poldmaa, L. Saag, I. Saar, A. Schussler, J. A. Scott, C. Senes, M. E. Smith, A. Suija, D. L. Taylor, M. T. Telleria, M. Weiss, and K. H. Larsson. "Towards a Unified Paradigm for Sequence-Based Identification of Fungi." Mol Ecol 22, no. 21 (2013): 5271-7.
  10. Edgar, R. C. "Accuracy of Taxonomy Prediction for 16s Rrna and Fungal Its Sequences." PeerJ 6 (2018): e4652.
  11. Taylor, D. Lee, A. Walters William, J. Lennon Niall, James Bochicchio, Andrew Krohn, J. Gregory Caporaso, and Taina Pennanen. "Accurate Estimation of Fungal Diversity and Abundance through Improved Lineage-Specific Primers Optimized for Illumina Amplicon Sequencing." Applied and Environmental Microbiology 82, no. 24 (2016): 7217-26.
  12. Tedersoo, Leho, Sten Anslan, Mohammad Bahram, Sergei Põlme, Taavi Riit, Ingrid Liiv, Urmas Kõljalg, Veljo Kisand, Henrik Nilsson, Falk Hildebrand, Peer Bork, and Kessy Abarenkov. "Shotgun Metagenomes and Multiple Primer Pair-Barcode Combinations of Amplicons Reveal Biases in Metabarcoding Analyses of Fungi." MycoKeys 10 (2015): 1-43.
  13. Ren, Chengjie, Zhenghu Zhou, Yaoxin Guo, Gaihe Yang, Fazhu Zhao, Gehong Wei, Xinhui Han, Lun Feng, Yongzhong Feng, and Guangxin Ren. "Contrasting Patterns of Microbial Community and Enzyme Activity between Rhizosphere and Bulk Soil Along an Elevation Gradient." Catena 196 (2021).
  14. Cui, Yongxing, Haijian Bing, Linchuan Fang, Yanhong Wu, Jialuo Yu, Guoting Shen, Mao Jiang, Xia Wang, and Xingchang Zhang. "Diversity Patterns of the Rhizosphere and Bulk Soil Microbial Communities Along an Altitudinal Gradient in an Alpine Ecosystem of the Eastern Tibetan Plateau." Geoderma 338 (2019): 118-27.

Reviewer 2 Report

The reviewed work is interesting and may constitute the basis for further research. The Authors present possible factors influencing the diversity of fungi. The research has not only theoretical but also practical value related to environmental protection.

I'm wondering where these differences in fungal diversity come from. The explanations provided by the Authors are convincing. However, it is worth considering the species structure of vegetation and devoting more space to it (as the Authors point out). In the case of Mt. In Seorak, the species composition of the vegetation includes birch. It is known that birches are considered engineering species, significantly influencing soil properties.  

The Authors demonstrated a significant impact of phosphorus on fungal communities. Bhalla et al. 2022 (The phosphate tongue of fungi Kabir Bhalla, Xianya Qu, Matthias Kretschmer, James W. Kronstad; Trends Microbiol. 2022 April ; 30(4): 338–349. doi:10.1016/j.tim.2021.08.002.) found significant the importance of phosphorus compounds influencing not only the development of fungi, but also their relationships with other organisms, especially bacteria. The discussion lacked references to the relationship between fungi and bacteria. I realize that the object of the research were fungi, but the problem could be discussed based on the literature.

Author Response

We express our profound gratitude to the editor and the reviewers for dedicating their valuable time to review our manuscript. We have made many corrections and edits throughout the manuscript. To facilitate an efficient review process, we have included all the comments within this response letter, ensuring ease of reference and comprehensive understanding.

Reviewer(s)' Comments to Author:

The major comments:

The reviewed work is interesting and may constitute the basis for further research. The Authors present possible factors influencing the diversity of fungi. The research has not only theoretical but also practical value related to environmental protection.

We appreciate your positive feedback on our research.

The detail comments

1. I'm wondering where these differences in fungal diversity come from. The explanations provided by the Authors are convincing. However, it is worth considering the species structure of vegetation and devoting more space to it (as the Authors point out). In the case of Mt. In Seorak, the species composition of the vegetation includes birch. It is known that birches are considered engineering species, significantly influencing soil properties.

We appreciate and fully agree with the reviewer's comment. We recognize the significant impact that aboveground communities, particularly plants, have on belowground communities, and indeed, we plan to further our research in the future by elucidating the effects of vegetation diversity and structure on the belowground communities in mountain forests on the Korean Peninsula. However, in this study, we aimed to address the simple but fundamental ecological question regarding the biodiversity patterns of fungal communities along elevational gradients and their potential influencing factors. To focus on this question, we sampled bulk soil rather than rhizosphere soil, thereby minimizing the direct influence of vegetation on fungal communities. While this approach may not completely exclude the effects of vegetation, it is considered applicable for exploring a broader range of environmental factors beyond vegetation[1-4]. Also, in the revised manuscript, we have included a brief discussion on the impact of birch trees (Betula spp.) on soil fungal diversity as follows (L355-L359).

2. The Authors demonstrated a significant impact of phosphorus on fungal communities. Bhalla et al. 2022 (The phosphate tongue of fungi Kabir Bhalla, Xianya Qu, Matthias Kretschmer, James W. Kronstad; Trends Microbiol. 2022 April; 30(4): 338–349. doi:10.1016/j.tim.2021.08.002.) found significant the importance of phosphorus compounds influencing not only the development of fungi, but also their relationships with other organisms, especially bacteria. The discussion lacked references to the relationship between fungi and bacteria. I realize that the object of the research were fungi, but the problem could be discussed based on the literature.

We sincerely appreciate the reviewer's suggestion to include the important reference in our work. According to Bhalla et al.[5], phosphate is crucial for fungal proliferation, stress responses, cell wall synthesis, and carbon metabolism, serving as a key mediator in interactions with other organisms. Its regulation affects fungal growth, diversity, and functionality across various environments. Phosphate availability influences fungal adaptation to environmental stress and host interactions, directly impacting the structure and diversity of soil fungal communities. Additionally, phosphate acts as a signaling molecule in fungal cells, regulating nutrient acquisition, stress responses, and metabolic processes, thereby enabling fungi to adjust their diversity and distribution in different soil environments. We have now revised the discussion section accordingly in the revised manuscript (L390-L397).

  1. Ren, Chengjie, Zhenghu Zhou, Yaoxin Guo, Gaihe Yang, Fazhu Zhao, Gehong Wei, Xinhui Han, Lun Feng, Yongzhong Feng, and Guangxin Ren. "Contrasting Patterns of Microbial Community and Enzyme Activity between Rhizosphere and Bulk Soil Along an Elevation Gradient." Catena 196 (2021).
  2. Shen, C., A. Gunina, Y. Luo, J. Wang, J. Z. He, Y. Kuzyakov, A. Hemp, A. T. Classen, and Y. Ge. "Contrasting Patterns and Drivers of Soil Bacterial and Fungal Diversity across a Mountain Gradient." Environ Microbiol 22, no. 8 (2020): 3287-301.
  3. Cui, Yongxing, Haijian Bing, Linchuan Fang, Yanhong Wu, Jialuo Yu, Guoting Shen, Mao Jiang, Xia Wang, and Xingchang Zhang. "Diversity Patterns of the Rhizosphere and Bulk Soil Microbial Communities Along an Altitudinal Gradient in an Alpine Ecosystem of the Eastern Tibetan Plateau." Geoderma 338 (2019): 118-27.
  4. Liu, D., G. Liu, L. Chen, J. Wang, and L. Zhang. "Soil Ph Determines Fungal Diversity Along an Elevation Gradient in Southwestern China." Sci China Life Sci 61, no. 6 (2018): 718-26.
  5. Bhalla, K., X. Qu, M. Kretschmer, and J. W. Kronstad. "The Phosphate Language of Fungi." Trends Microbiol 30, no. 4 (2022): 338-49.

Round 2

Reviewer 1 Report

Great care by the authors is evident from the revisions made. Most of them are at least adequate, but two points remain:

1. Authors' responses to points 1.1, 1.2 and 1.3 (Major comments)

I fully understand what are the main questions of this study. However, in my opinion, bringing more attention to the five-fold replication level, which is explicitly (in statistical terms) the variation residual, is a formal necessity. Your interpretation that it is due to differences in the local biota does not seem likely, assuming that this factor does not change much over the 4 x 20m distance taken by each replicate set. This is why I feel that the new text in lines 410-420 does not hit the mark. The residual in your study is probably due to the well-known spatially "messy" variation in the composition of the fungal communities, and that is why I suggested the Bray-Curtis nMDS to depict the consistency between samples of the same five-fold replicate set.

2. Authors' response to point 16 (Detail comments)

The question I raised concerns now lines 441-450, and I maintain that this is a biologically unsound discussion.

Lines 174-176

This is just a remark. We can all agree that the 97% threshold has been "widely used", indeed! But it does not mean that this is the adequate choice. Take for example reference 45: regarding fungal data, the species level comes at >98% (dataset F, https://doi.org/10.7717/peerj.4652/supp-6). Reference 46 does not provide any support for the 97% threshold choice, and reference 47 actually cites (in the discussion) the assertion by Kõljalg et al. (2013) that 97% is too low (I am surprised by its exclusion from the reference list, since you are obviously aware of that paper).

However, the text is much improved now in justifying the choice of this threshold and, in spite of my reservations on the 97% level in itself, this improvement renders the text in question very much acceptable: for the purpose of your study, 97% can be fine.

Line 187

I suggest, for clarity, changing «Unclassified OTUs were categorized as "Others" (Figure. S2)» to «Otherwise, OTUs remained unclassified and categorized as "Others" (Figure. S2)».

Line 189

"we have revised the manuscript to use the terms" is factual, but seems inadequate for the final publication. I suggest "we adopt the terms".

Lines 318-319

I suggest -- to avoid the repetition with "exhibiting": "Species diversity varies along elevational gradients, with fungal communities often exhibiting particularly complex patterns".

Author Response

We are grateful for your time and effort in providing such valuable feedback. Your constructive comments and guidance have not only improved the current manuscript but have also provided us with insights that will benefit our future research endeavors. We have carefully addressed all comments and made the necessary revisions in the manuscript. We believe these changes have strengthened our study and we look forward to any further feedback.

Major comments

Great care by the authors is evident from the revisions made. Most of them are at least adequate, but two points remain:

1. Authors' responses to points 1.1, 1.2 and 1.3 (Major comments)

I fully understand what are the main questions of this study. However, in my opinion, bringing more attention to the five-fold replication level, which is explicitly (in statistical terms) the variation residual, is a formal necessity. Your interpretation that it is due to differences in the local biota does not seem likely, assuming that this factor does not change much over the 4 x 20m distance taken by each replicate set. This is why I feel that the new text in lines 410-420 does not hit the mark. The residual in your study is probably due to the well-known spatially "messy" variation in the composition of the fungal communities, and that is why I suggested the Bray-Curtis nMDS to depict the consistency between samples of the same five-fold replicate set.

We appreciate your insightful comments and agree that our initial explanation did not adequately address the residual variation observed in the five-fold replication at the same elevation. To validate the consistency of samples within the same elevation, we have conducted an analysis with nMDS as recommended. The results of the nMDS analysis clearly confirmed the consistency of samples within each elevation band (replicate set). Additionally, we conducted a clustering analysis, which supported the nMDS results, indicating that samples from the same elevation band within the same mountain tend to cluster together more closely than samples from different elevation bands. We have now revised the manuscript to provide further clarification in the Methods section and have modified the Discussion section to enhance the clarity of the manuscript. (L139-L142; L326-L330; Figure S1)

2. Authors' response to point 16 (Detail comments)

The question I raised concerns now lines 441-450, and I maintain that this is a biologically unsound discussion.

We have carefully reconsidered the content in question and agree with the reviewer's perspective. We have removed this part of the discussion from the manuscript to avoid any over-interpretation. (L442)

Detail comments

Lines 174-176

This is just a remark. We can all agree that the 97% threshold has been "widely used", indeed! But it does not mean that this is the adequate choice. Take for example reference 45: regarding fungal data, the species level comes at >98% (dataset F, https://doi.org/10.7717/peerj.4652/supp-6). Reference 46 does not provide any support for the 97% threshold choice, and reference 47 actually cites (in the discussion) the assertion by Kõljalg et al. (2013) that 97% is too low (I am surprised by its exclusion from the reference list, since you are obviously aware of that paper).

However, the text is much improved now in justifying the choice of this threshold and, in spite of my reservations on the 97% level in itself, this improvement renders the text in question very much acceptable: for the purpose of your study, 97% can be fine.

Thank you very much for the valuable insights on the choice of the 97% threshold for OTU clustering. Your comments have helped us improve the manuscript, and we are glad that the justification for our chosen threshold is now acceptable for the purpose of our study.

Line 187

I suggest, for clarity, changing «Unclassified OTUs were categorized as "Others" (Figure. S2)» to «Otherwise, OTUs remained unclassified and categorized as "Others" (Figure. S2)».

We appreciate your suggestion for improving the clarity of our manuscript. We have revised the text in the updated manuscript as recommended. (L190-L191)

Line 189

"we have revised the manuscript to use the terms" is factual, but seems inadequate for the final publication. I suggest "we adopt the terms".

Thank you for pointing out this issue. We have revised the text in the updated manuscript as recommended. (L192)

Lines 318-319

I suggest -- to avoid the repetition with "exhibiting": "Species diversity varies along elevational gradients, with fungal communities often exhibiting particularly complex patterns".

We appreciate your suggestion for improving the clarity of our manuscript. We have revised the text in the updated manuscript as recommended. (L321-L322)
